# Delivering an Immunocastration Vaccine via a Novel Subcutaneous Implant

**DOI:** 10.3390/ani12192698

**Published:** 2022-10-07

**Authors:** Andrew K. Curtis, Douglas E. Jones, Michael Kleinhenz, Shawnee Montgomery, Miriam Martin, Mikaela Weeder, Alyssa Leslie, Balaji Narasimhan, Sean Kelly, Drew R. Magstadt, Alfredo Colina, Johann F. Coetzee

**Affiliations:** 1Department of Anatomy and Physiology, College of Veterinary Medicine, Kansas State University, Manhattan, KS 66506, USA; 2Department of Veterinary Pathology, College of Veterinary Medicine, Iowa State University, Ames, IA 50011, USA; 3Department of Clinical Sciences, College of Veterinary Medicine, Kansas State University, Manhattan, KS 66506, USA; 4Department of Chemical and Biological Engineering, Nanovaccine Institute, Iowa State University, Ames, IA 50011, USA; 5Department of Veterinary Diagnostic and Production Animal Medicine, College of Veterinary Medicine, Iowa State University, Ames, IA 50011, USA; 6Department of Microbiology and Immunology, Medical College of Wisconsin, Milwaukee, WI 53226, USA

**Keywords:** vaccine, castration, dairy cattle, immunocastration, animal welfare, refinement, Holstein, implant

## Abstract

**Simple Summary:**

Male cattle (bulls) are often castrated as part of routine herd management. The benefits of this practice include the reduction of aggression and the elimination of unwanted pregnancies. However, castration represents an animal welfare concern as bulls are subjected to pain during and after the procedure. Surgical castration, in particular, places animals at increased risk of hemorrhage and infection. Immunocastration, a method involving vaccination against the hormones that regulate reproduction, offers a reduced-pain alternative to traditional castration, but the current products require multiple doses to effectively reduce fertility for extended periods. In an effort to improve upon current multi-dose immunocastration strategies, we evaluated the efficacy of a single-dose implantable immunocastration vaccine. This implant was designed to reduce fertility without the need for multiple doses, thus improving welfare for the animals as well as safety for producers and clinicians. The results presented here are promising and suggest that further refinement of the immunocastration implant could provide a convenient alternative to current immunocastration strategies.

**Abstract:**

Immunocastration relies on the vaccine-mediated stimulation of an immune response to gonadotropin-releasing hormone (GnRH) in order to interrupt spermatogenesis. This approach offers a less painful alternative to traditional castration approaches but the current, commercially available options require multiple doses of vaccine to maintain sterility. Thus, a series of pilot studies were conducted to determine the feasibility of a single-dose immunocastration vaccine implant. These five studies utilized a total of 44 Holstein bulls to determine the optimal vaccine composition and validate the ability of a stainless-steel subcutaneous implant to deliver a vaccine. Outcome measures included the duration of implant retention, scrotal dimensions and temperature, implant site temperature, anti-GnRH antibodies, and serum testosterone concentration. Over the course of several studies, anti-GnRH antibodies were successfully stimulated by vaccine implants. No significant treatment effects on scrotal dimensions or testosterone were detected over time, but changes in spermatogenesis were detected across treatment groups. Results indicate that a single-dose implantable immunocastration vaccine elicits a humoral immune response and could impact spermatogenesis in bulls. These findings provide opportunities for the refinement of this technology to improve implant retention over longer periods of time. Taken together, this approach will offer producers and veterinarians an alternative to physical castration methods, to improve animal welfare during routine livestock management procedures.

## 1. Introduction

Approximately 88% of male beef cattle are castrated in the United States [1]. This translates to around 17 million procedures per year, making it one of the most common livestock management practices currently employed by the beef cattle industry [2]. Cattle are castrated for a variety of reasons that generally pertain to increased ease of management and improved carcass traits. Castrates exhibit reduced aggressiveness [3], reduced mounting behavior and mounting-related injuries [4], and improved meat quality and market premium [5,6]. Importantly, the sterilization that results from castration prevents unwanted breeding [7].

Methods of castration are varied, but all have side effects and cause the animal pain [7]. The American Veterinary Medical Association’s Animal Welfare Division divides the forms of castration between physical, chemical, and immunological methods (i.e., immunocastration [8]). Physical castration involves the surgical removal of testicles, the application of a constricting elastic band at the base of the scrotum, and/or external clamping such as a Burdizzo clamp [9]. Physical castration predominates in production settings and is commonly performed in combination with other painful husbandry practices, such as dehorning and branding [10,11,12]. Chemical castration includes the injection of sclerosing or toxic agents into the testicular parenchyma to cause irreparable damage and loss of function [8], but this may preserve androgenesis and the associated behavior [13].

A less painful alternative to current physical castration approaches relies on immunization against reproductive hormones to control function (immunocastration). One target for vaccine development has been the gonadotropin-releasing hormone (GnRH) for its upstream role in the relevant endocrine signaling cascades. Vaccination against GnRH conjugated to carrier proteins (i.e., hapten-carrier complexes) such as human serum albumin [14] and ovalbumin (OVA) [15,16], has been shown to provoke humoral immune responses in bulls and heifers. Early work with rams and bull calves relied on 4 doses of vaccine to immunize against GnRH and resulted in diminished testis size and weight, as well as reduced plasma testosterone concentrations [17]. Since 2007, one vaccine (Bopriva^®^, Zoetis, Parsippany, NJ, USA) marketed specifically for use with cattle has been available in several markets outside the USA. Janett et al. [18] reported a significant reduction in testosterone levels, testicular development, and physical activity in pubertal bulls treated with Bopriva^®^. However, studies examining Bopriva^®^ and other immunocastration vaccine formulations have relied on multiple doses of vaccine in order to impair reproductive function in cattle [18,19,20]. A GnRH vaccine that replicates the benefits of surgical castration with a single dose would be safer and more convenient for producers and clinicians interested in reducing pain during routine management. Specifically, a single-dose solution would require fewer animal-handling events and an implantable device would eliminate the risk of needlestick injuries to handlers. To this end, a series of proof-of-concept studies were conducted in order to test the retention of an implantable vaccine and determine its ability to stimulate humoral immunity, limit testicular development, and reduce testosterone production in bulls.

## 2. Materials and Methods

All experimental procedures were reviewed and approved by the Institution Animal Care and Use Committee (protocol #4394) at Kansas State University. The study animals were all healthy male dairy cattle and ranged from 3 to 14 months of age at the time of enrollment.

### 2.1. Vaccine Design

The primer doses (used in pilot studies 1–4) were soluble injections, whereas the boost and VPEAR were solid components that were designed to be released from the implant. Stainless-steel cylindrical implants measured approximately 5 mm × 41 mm and were delivered through a standard Compudose^®^ (Elanco Animal Health, Greenfield, IN, USA) needle, using a proprietary applicator. Cylinders were packed with vaccine components and sealed at one end with a 0.65-micron porous hydrophilic polyvinylidene fluoride (PVDF) membrane (Durapore^®^, Millipore Sigma, Burlington, MA, USA). Dry mixtures of all vaccine components were pressed in a custom-made mold at 0.5 tons-on-ram for 5 s, using a hydraulic press (International Crystal Laboratories Inc., Garfield, NJ, USA). The implant was designed and formulated in a way similar to that previously described [21]. Depending on iteration, GnRH was complexed as a hapten, with either OVA or keyhole limpet hemocyanin (KLH) carrier proteins. Vaccine ingredients were arranged sequentially for the controlled delivery of both antigens, as well as between 0.5 and 100 mg of the adjuvants diethylaminoethyl-dextran (DEAE-D, Sigma-Aldrich, St. Louis, MO, USA) and/or Quil-A (InvivoGen, San Diego, CA, USA). Implant design was such that a priming component (primer) was presented first (Pilot 1), then a boost component, then the vaccine platform for extended antigen release (VPEAR), adapted from the work of Boggiatto et al. [22]. All implants contained between 0.375 and 3.9 mg trehalose, as well as between 140 and 200 mg polyanhydride.

### 2.2. Implantation

Prior to implantation, hair was removed from the caudal aspects of the ears using livestock clippers (Powerpro^®^ Ultra, Oster, Milwaukee, WI, USA) and skin was cleaned using chlorhexidine surgical scrub (Chlorhexidine 4%, VetOne, Boise, ID, USA) and gauze soaked in 70% isopropyl alcohol (Vedco, St. Joseph, MO, USA). A local nerve block was provided using injections of lidocaine hydrochloride (without epinephrine; Lidocaine 2%, VetOne, Boise, ID, USA) approximately 5 min before implantation. Implants were placed in the subcutaneous space of the caudal aspect of the ear pinna. The incisions were approximately 0.5 cm long and were sealed using a single suture (000 PDS, Ethicon, Raritan, NJ, USA) and cyanoacrylate (Loctite^®^ Super Glue, Henkel North American Consumer Goods, Hartford, CT, USA).

### 2.3. Blood Sampling and Analysis

During the pilot studies that relied on blood draws, the sampling regimen was performed every 14 days. Beginning at day 0, immediately before vaccination, baseline blood was collected from the jugular vein into evacuated tubes (Vacutainer^®^, Becton Dickinson, Franklin Lakes, NJ, USA) containing ethylenediaminetetraacetic acid (1.8 mg/mL whole blood) or no anticoagulant. Approximately 10 mL of whole blood was drawn each time. Blood was centrifuged (IEC Centra^®^ CL2, Thermo Electron Corporation, Waltham, MA, USA) at 1150× *g* for 10 min, the serum and plasma were drawn off, and samples were frozen in cryotubes at −27 °C until further analysis. A commercially available double-antibody radioimmunoassay (RIA) kit was used to detect the total unconjugated testosterone (125 I RIA Kit, MP Biomedicals LLC, Solon, OH, USA) according to the manufacturer’s instructions. Testosterone was then measured via RIA, using an automatic gamma counter (2470 Wizard2^®^, PerkinElmer, Waltham, MA, USA). Anti-GnRH antibodies were assessed at 1:10,000 dilution via enzyme-linked immunosorbent assays (ELISA), using EvenCoat^®^ streptavidin-coated plates (Cat# P004, R&D Systems, Minneapolis, MN, USA). The optical density (OD) was measured using a SpectraMax^®^ i3 Multi-Mode Microplate Reader (Molecular Devices LLC, San Jose, CA, USA).

### 2.4. Scrotal and Ear Surface Temperature

Infrared thermography (IRT) was used to better monitor the inflammatory responses to vaccination during the study. The temperature of the caudal scrotal surface was assessed via IRT, using a digital camera capable of capturing thermographic images (TiX580 Thermal Imager, Fluke Corporation, Everett, WA, USA). The thermographic camera was perpendicularly positioned approximately 45 cm away from the caudal aspect of the surface of the scrotum; image focus and quality were verified before saving the image to memory. Similarly, images of the implant site were captured by aiming the camera at the caudal aspect of the ear from approximately 20 cm away. Scrotal and ear surface temperatures were assessed via thermography every 14 days. Images were evaluated using the Fluke Tools software (Smartview^®^ 4.3, Fluke Corporation, Everett, WA, USA).

### 2.5. Scrotal and Testicular Dimensions

Scrotal dimensions were assessed using a flexible scrotal tape (Reliabull^®^, Lane Manufacturing Inc., Denver, CO, USA) and digital calipers (Tool Shop^®^ 6″ stainless steel digital caliper, Menards, Eau Claire, WI, USA). To measure the scrotal circumference (mm), both testicles were manipulated by hand so that they rested at the lowest and most distal aspect of the scrotum. The scrotum was then held firmly with one hand while the scrotal tape was applied at the level judged to have the largest circumference. The tape was then drawn firmly against the circumference of the scrotum to provide a value, as per the manufacturer’s instructions. The volume of each testicle was estimated by measuring the external length, width, and depth of individual testes through the scrotal skin, using digital calipers. Approximate volume (cc) was then calculated from the caliper measurements, using the prolate spheroid formula [23]. From this value, testicular mass (g) was also estimated in vivo.

### 2.6. Surgical Castration

In cases where animals were surgically castrated, animals were provided with systemic pain management in the form of oral meloxicam. There are no US FDA-approved formulations of meloxicam for use in cattle. Use in this case was approved by the IACUC as part of a valid veterinary-client-patient relationship. At the time of castration, animals were bolused with 15 mg tablets of meloxicam at a rate of 1 mg/kg bodyweight. Scrotums were washed using 4% chlorhexidine (VetOne) and gauze soaked in 70% isopropyl alcohol (Vedco). Local anesthesia was induced by injecting 5 mL of 2% lidocaine hydrochloride (without epinephrine) into each spermatic cord (VetOne). Open orchidectomy was performed approximately five minutes after applying the nerve block. Animals were monitored under close surveillance for 2 h and then once daily for 14 days, as per institutional requirements. Harvested testes were immediately placed into 10% buffered formalin for histological evaluation.

### 2.7. Testes Histology

In those cases where animals were surgically castrated, the testes were histologically evaluated to compare the differences in spermatogenesis between treatment groups. Histologic slides were prepared, and the analysis used 3 parenchymal slides and 1 epididymis from each testicle. Seminiferous tubular cross-sections were selected at random (6 or 7 per slide) for each testicle. Scoring relied on methods that have previously been described by Johnsen [24] and adapted by Daigle et al. [25], with higher numbers indicating greater degrees of spermatogenesis (Table 1). Scoring was conducted independently by the Veterinary Diagnostic Laboratory at the Iowa State University College of Veterinary Medicine and the scorers were blinded to treatment.

### 2.8. Statistical Analysis

Statistical analyses evaluated the relationships between treatment and scrotal dimensions (in mm, g, or cc), antibody production, and testosterone over time. The treatment effect on spermatogenesis among castrates was also examined. The scrotal dimensions, antibodies, and testosterone concentration over time were analyzed as repeated measures using JMP^®^ (SAS Institute, Cary, NC, USA). The treatment’s effect on spermatogenesis, quantified as described by Daigle et al. [25], was determined using an unpaired Student’s *t*-test. For all outcomes, the individual animal was considered the experimental unit, and statistical significance was set a priori, at *p* < 0.05. Implant retention was confirmed by palpation at each sampling timepoint but was not analyzed as an outcome. No direct comparisons between pre- and post-pubertal groups were made.

### 2.9. Treatments

The treatments that were given are summarized in Table 2. Five pilot trials were studied and are outlined in the following sections.

#### 2.9.1. Pilot 1

A total of 11 dairy bull calves (aged 6 months and with fully descended testicles) were randomly assigned to one of four treatment groups (*n* = 2–3) using the RAND function in a spreadsheet program (Excel^®^, Microsoft Corporation, Richmond, WA, USA). All animals received a soluble vaccine primer containing 100 mg polyvinylpyrrolidone (PVP), 100 mg DEAE-D, and 0.25 mg GnRH linked to KLH (GnRH-KLH) for a total of 0.1 mg GnRH. Briefly, implantable vaccine constructs consisted of 1 out of 4 GnRH-based treatment iterations (including a boost and VPEAR) with varying amounts and types of adjuvants. The treatments included GnRH, complexed with OVA (GnRH-OVA) and KLH (GnRH-KLH), in addition to the adjuvants DEAE-D and Quil-A^®^ (Implants 1 and 2) and/or just DEAE-D (Implants 3 and 4). All implants contained a total of 1.2 mg GnRH peptide and 100 mg of PVP. The calves were sampled every 2 weeks for a total of 56 days. At the end of the study, the calves were surgically castrated, and the gonads were histologically evaluated. The major outcome measures for Pilot 1 were external scrotal/testicular changes measured over time, as well as the registration of spermatogenesis by histology following castration.

#### 2.9.2. Pilot 2

In total, 12 male Holstein calves (aged 8 months) were implanted with empty stainless-steel implants (Implant 5) in order to determine the viability of stainless steel as a biocompatible delivery system. Calves were monitored for 42 days, and implant-site reactions were monitored, along with attrition. The primary outcome measure for Pilot 2 was implant rejection over time. Rejection was defined as a total loss of the device and its contents from the subcutaneous space.

#### 2.9.3. Pilot 3

A total of 12 male Holstein calves (aged 10 months) were randomly allocated to 1 of 2 treatment groups. Group 1 (*n* = 6) received Implant 6 (boost only) and Group 2 (*n* = 6) received Implant 7 (VPEAR only). Animals were monitored for 175 days and sampled every 14 days. Implant 6 contained GnRH-KLH with no adjuvant, whereas Implant 7 contained GnRH-OVA with DEAE-D and Quil-A^®^. Outcome measures included implant attrition and anti-GnRH antibody production, as assessed by ELISA.

#### 2.9.4. Pilot 4

A total of 8 male Holstein calves (aged 14 months) were administered Implant 8 (boost and VPEAR). Implant 8 contained GnRH-OVA and GnRH-KLH with both Quil-A^®^ and DEAE-D. Animals were monitored for 56 days and sampled every 14 days. Outcome measures included implant attrition and anti-GnRH antibody production, as assessed by ELISA.

#### 2.9.5. Pilot 5

A total of 12 male Holsteins (aged 3 months, with fully descended testicles) were enrolled in a study to examine two implantable vaccines (Implant 9 and Implant 10, boost and VPEAR). Implant 9 (*n* = 6) contained 1.1 mg of GnRH antigen, whereas Implant 10 (*n* = 6) served as a negative control and contained only scrambled peptide (SP), linked to KLH (SP-KLH) and OVA (SP-OVA). The SP implants delivered a total of 1.1 mg of the constituent amino acids of GnRH in a randomized sequence. Animals were monitored for 252 days and sampled every 14 days. Outcome measures included rectal temperature, body weight (BW), scrotal circumference, estimated testicle volume, a histological evaluation of spermatogenesis, temperature changes at the scrotum and implant site, and testosterone concentrations.

## 3. Results

### 3.1. Pilot 1

During the course of the study, five out of eleven implants were rejected before day 28. Within 6 weeks, all implants had been rejected. At the study’s conclusion, animals were surgically castrated, and the testes were examined. Post-castration histology scoring revealed significant differences (*p* < 0.05) in the extent of spermatogenesis among treatment groups (Figure 1 and Figure 2). Animals provided with Implant 3 exhibited significantly (*p* < 0.0001) less spermatogenesis than animals treated with Implants 1, 2, and 4. Animals treated with Implant 2 exhibited significantly (*p* < 0.0001) less spermatogenesis than those treated with 1 and 4, but significantly (*p* < 0.0001) more than those with Implant 3. There was no significant difference (*p* = 0.5186) in spermatogenesis noted between animals treated with Implants 1 and 4. Scrotal measurements were also compared across treatment groups. There was no significant treatment by time effect on total estimated testicular volume (*p* = 0.9387), total estimated testicular mass (*p* = 0.9387), scrotal circumference (*p* = 0.9934), or percentage change in circumference over the baseline measurements (*p* = 0.0809).

### 3.2. Pilot 2

During the course of the study, empty stainless-steel implants were monitored for rejection. Of the twelve calves that received Implant 5, only two lost their implants over the course of 42 days. One implant was rejected between days 7 and 14, while the other was rejected between days 35 and 42.

### 3.3. Pilot 3

Anti-GnRH antibody responses were measured using an ELISA. A total of five of the twelve implants were rejected over the first 105 days. Over the course of the study, the antibody response was significantly greater (*p* < 0.0001) for animals administered Implant 7 than for those administered Implant 6. Additionally, there was a significant (*p* < 0.0001) treatment by time interaction. Antibody levels among animals administered Implant 7 were significantly higher than in animals administered Implant 6 at day 14 (*p* = 0.0001), 28 (*p* = 0.0001), and 91 (*p* = 0.0140). Interestingly, at day 161, antibody levels were significantly higher (*p* = 0.0461) among those animals administered Implant 6 than in those animals administered Implant 7 (Figure 3).

### 3.4. Pilot 4

The anti-GnRH antibody responses were measured using an ELISA. Over the course of the study, half of the animals administered Implant 8 rejected the implant. Animals administered Implant 7 had significantly higher antibody levels when compared to animals administered Implant 8 (*p* = 0.0134). Animals administered Implant 8 also had significantly (*p* < 0.0001) higher antibody levels than animals administered Implant 6. In terms of antibodies, there was a significant (*p* = 0.0044) treatment by time interaction. Antibody levels among animals administered Implant 8 were significantly higher than animals administered Implant 6 at days 14 (*p* < 0.0001) and 28 (*p* = 0.0015). On day 28, animals administered Implant 7 displayed significantly elevated (*p* = 0.0294) antibody responses when compared to animals administered Implant 8 (Figure 4).

### 3.5. Pilot 5

Over the course of the study, all but three implants were rejected. However, there was a significant (*p* = 0.0051) antibody difference between treatment groups, with those animals receiving Implant 9 exhibiting a greater antibody response than animals that received Implant 10 (Figure 5). However, there was no treatment by time interaction (*p* = 0.1002). There were no significant time by treatment interactions regarding raw scrotal circumference (*p* = 0.815), percentage change over baseline (*p* = 0.8315), or estimated testicular volume (*p* = 0.3677). Likewise, there was no treatment effect over time on BW (*p* = 0.9977), average scrotal temperature (*p* = 0.6766), or average implant site temperature (*p* = 0.9137). Finally, there was no treatment by time effect on testosterone concentrations between groups (*p* = 0.2574). Once the study had concluded, the animals were surgically castrated, and the testes were examined. Post-castration histology scoring revealed no significant (*p* = 0.4917) differences in terms of the extent of spermatogenesis between treatment groups.

## 4. Discussion

The histological results from Pilot 1 support the use of the adjuvant DEAE-D in immunocastration vaccine design. Animals treated with a vaccine containing 210 mg DEAE-D and no Quil-A^®^ exhibited reduced spermatogenesis when compared to animals treated with only 130 mg of DEAE-D. Similarly, animals treated with 210 mg DEAE-D exhibited less spermatogenesis than animals administered implants containing any amount of Quil-A^®^. Previous work has also indicated variable adjuvant effects on experimental GnRH vaccines. Studies examining oil-adjuvanted (Freund’s complete with Freund’s incomplete) GnRH-KLH vaccines showed that they gave poor suppression of testosterone [26] but appeared to reduce aggressive behavior [27]. Another experiment using GnRH-OVA in a water-oil adjuvant containing *Mycobacterium butyricum* was unsuccessful at reducing testosterone to castration levels but did appear to reduce testicular growth [28]. It is possible that different outcomes could be related to differences in adjuvants or other vaccine components, and further work is needed to fully describe these differences in the context of single-dose GnRH vaccines. Additionally, implant rejection may also have a limited duration and profundity in terms of the immune response.

The implant retention noted during Pilot 2 was in agreement with previous work that has demonstrated the biocompatibility of stainless steel in animal models [29]. As such, attrition during Pilot 1 was attributed to secondary infection. Successive pilots incorporated 0.5–1.0 mg oxytetracycline (OTC) in the implants, in an attempt to combat this.

The results from Pilot 3 suggest that the addition of at least some adjuvant is useful in stimulating a humoral anti-GnRH response. Implant 7 contained both DEAE-D and Quil-A^®^, whereas Implant 6 contained neither. This result is in agreement with previous work, including a recent experiment [30] that demonstrated significantly increased anti-GnRH antibodies in response to a recombinant GnRH vaccine adjuvanted with chitosan. Despite the addition of OTC, implant attrition could again have impacted the quality of the immune response to vaccination during Pilot 3.

Similar to those of Pilot 3, the results from Pilot 4 support the use of adjuvant(s) in the design of immunocastration vaccines. Antibody response to GnRH was elevated in adjuvanted implants when compared to Implant 6. It is conceivable that animal age had an impact on the humoral immune response, but as all animals were healthy and post-pubescent, this particular factor was not compared across studies. The use of DEAE-D is in agreement with previous work that has examined immunocastration vaccines in pigs (Improvac^®^; [31]) and cattle (Bopriva^®^; [32,33]). The use of the adjuvant Quil-A^®^ has been documented as a component of immunocastration products aimed at controlling estrus in horses and deer (Equity^®^; [34]). The results suggest that adjuvants are valuable components of an implantable immunocastration vaccine, but future work is needed to determine their impact on long-term implant retention.

Testicular measurements did not suggest a treatment effect in Pilot 5. Previous studies have reported inconsistent results from immunocastration in terms of the size of gonads. One study using an experimental GnRH-KLH vaccine with Freund’s complete and incomplete adjuvants noted a reduction in aggressive behavior, but no change in testicular weight in response to treatment [35]. Conversely, Cook et al. [36] found that GnRH-vaccinated animals showed significantly reduced scrotal circumference over time when compared to controls. In addition, BW did not appear to be affected by treatment during Pilot 5. This is in agreement with the findings of Amatayakul-Chantler et al. [37], who found no impact on BW in response to treatment with Bopriva^®^. Implant 9 appeared successful in stimulating anti-GnRH antibodies, but testosterone production remained unchanged. This is contrary to previous studies that showed an inverse relationship between anti-GnRH antibody production and androgenesis [38]. Ultimately, implant attrition was likely a factor in Pilot 5, as nine of the twelve implants were rejected over the course of the trial, despite the use of OTC. Predominant in the published literature are examples of testosterone suppression via soluble vaccines [39,40,41], as opposed to implants. Soluble vaccines, despite requiring boost doses, are not subject to rejection from the animal. It is plausible that treatment effects would have been more robust if implants had been retained throughout the experiment.

## 5. Conclusions

An immunocastration vaccine capable of delivering the benefits of surgical castration in a single dose remains an attractive prospect for producers, clinicians, and consumers interested in reducing pain during routine animal husbandry. The results presented here may serve as part of the foundation on which future implantable vaccines are built. Future work is needed to determine the optimal cocktail of ingredients required to increase implant retention, reduce testosterone, and halt spermatogenesis. Our results so far suggest the involvement of the humoral immune system, but future work should also examine the role that adaptive immunity plays in responding to vaccines delivered over an extended period of time via implants. Our use of stainless steel yielded variable lengths of retention. It is possible that other materials offer more attractive physical properties for a subcutaneous implant. Follow-up studies should prioritize the use of both vetted and novel biocompatible materials for the construction of the implant itself.

## 6. Patents

Vaccine Delivery Devices, US patent #10,130,454; Jones, Brewer, Narasimhan, Jackman. Assigned 20 November 2018.

## Figures and Tables

**Figure 1 animals-12-02698-f001:**
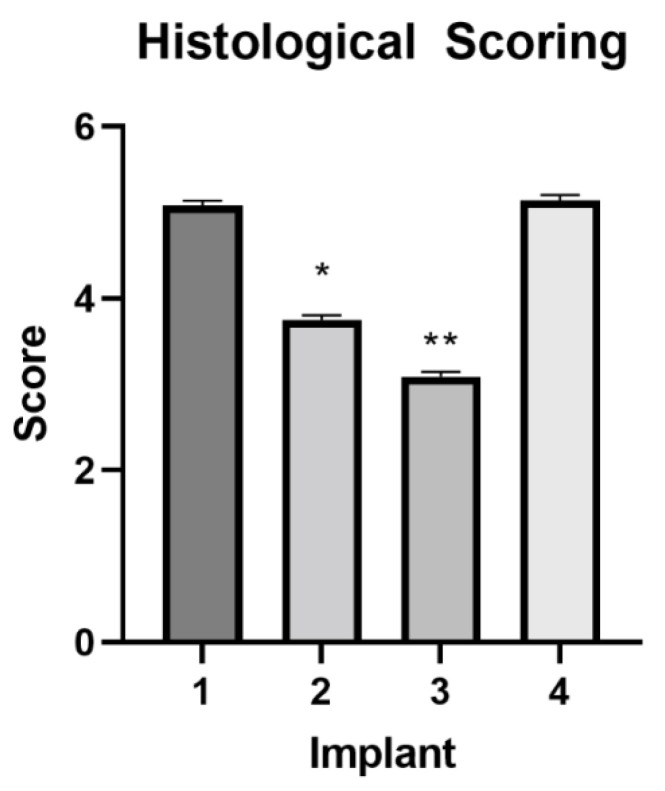
Differences in the extent of spermatogenesis among treatment groups in Pilot 1. Scores not connected by asterisks (* or **) are significantly different (*p* < 0.05).

**Figure 2 animals-12-02698-f002:**
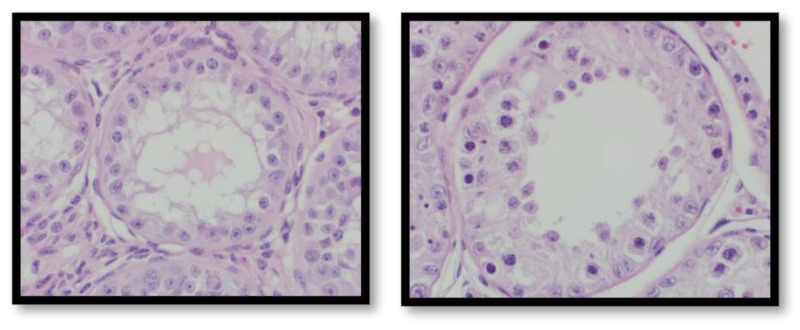
Representative examples of seminiferous tubules, showing an Ottesen score of 3 with only spermatogonia (**left**) and 6 with several spermatids (**right**).

**Figure 3 animals-12-02698-f003:**
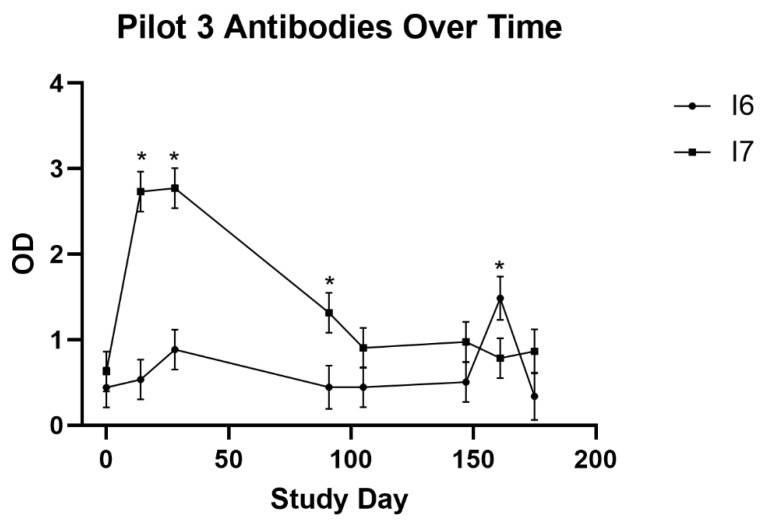
Antibody changes over time, as measured during Pilot 3. An asterisk (*) denotes significantly different (*p* < 0.05) optical densities (OD), indicative of antibody concentration, for that study day.

**Figure 4 animals-12-02698-f004:**
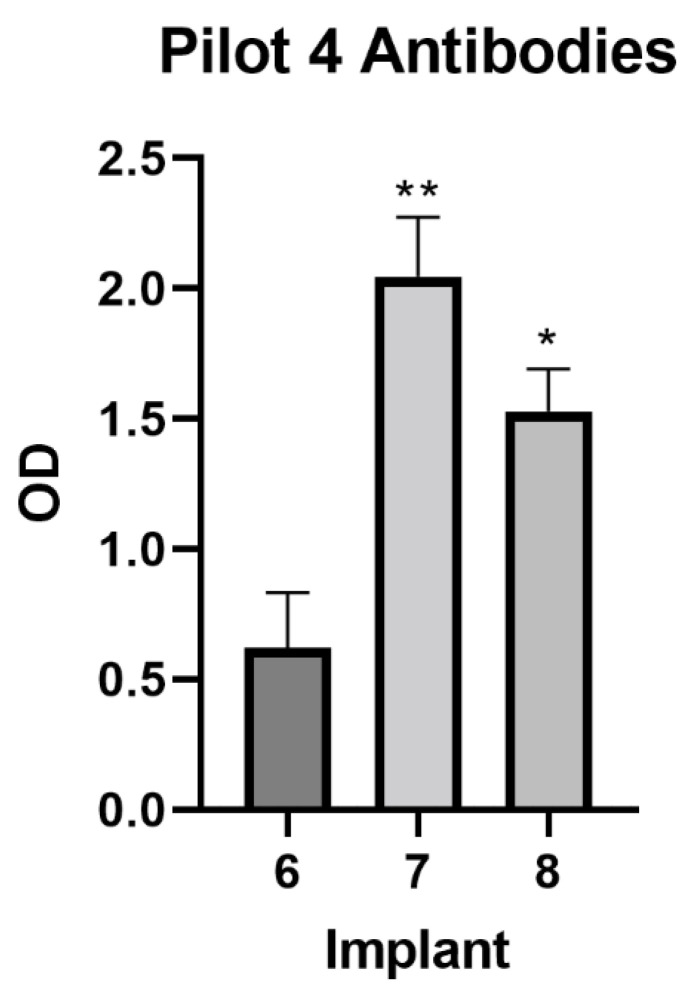
Differences in antibody production as measured on day 28 of Pilots 3 and 4. Implants not connected by asterisks (* or **) have significantly different (*p* < 0.05) optical densities (OD), indicative of antibody concentration.

**Figure 5 animals-12-02698-f005:**
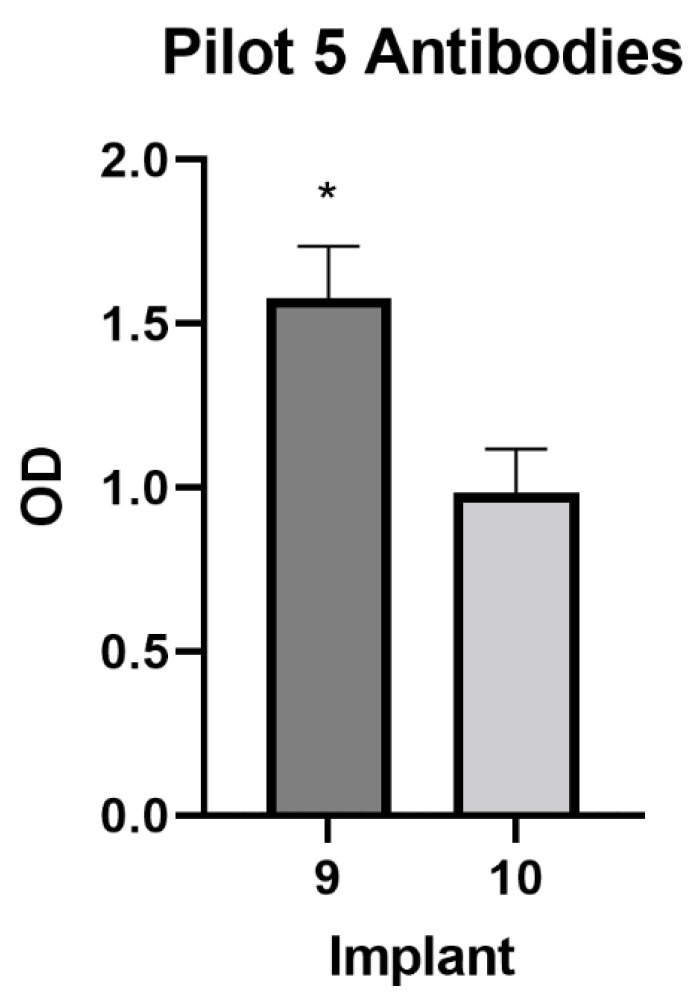
Differences in pooled antibody production, as measured over the course of Pilot 5. An asterisk (*) denotes a significant (*p* < 0.05) difference in optical density (OD), indicative of antibody concentration.

**Table 1 animals-12-02698-t001:** A scoring system, designed to quantify the degree of spermatogenesis.

Score	Description
1	No cells in tubular cross-section
2	Sertoli cells only
3	Spermatogonia only
4	No spermatozoa, no spermatids, <5 spermatocytes
5	No spermatozoa, no spermatids, many spermatocytes
6	No spermatozoa, <5–10 spermatids
7	No spermatozoa, many spermatids
8	All stages present, <5–10 spermatozoa
9	Many spermatozoa, germinal epithelium disorganized
10	Complete spermatogenesis

**Table 2 animals-12-02698-t002:** A summary of the implanted vaccines used in this study. All values are in milligrams. The duration is given in days.

Implant	Pilot	Duration	Stage	DEAE-D	Quil-A	GnRH-KLH	GnRH-Ova	Total GnRH	SP-KLH	SP-OVA	Total SP	OTC
1	1	56	Boost	100	5	0.25	-	0.1	-	-	-	-
			VPEAR	10	0.5	-	2.6	1	-	-	-	-
2	1	56	Boost	20	1	0.25	-	0.1	-	-	-	-
			VPEAR	10	0.5	-	2.6	1	-	-	-	-
3	1	56	Boost	100	-	0.25	-	0.1	-	-	-	-
			VPEAR	10	-	-	2.6	1	-	-	-	-
4	1	56	Boost	20	-	0.25	-	0.1	-	-	-	-
			VPEAR	10	-	-	2.6	1	-	-	-	-
5	2	42	Empty	-	-	-	-	-	-	-	-	-
(2)												
(2)												
6	3	175	Boost	-	-	0.25	-	0.1	-	-	-	0.5
			VPEAR	-	-	-	-	-	-	-	-	-
7	3	175	Boost	-	-	-	-	-	-	-	-	-
			VPEAR	10	0.5	-	2.6	1	-	-	-	0.5
8	4	56	Boost	-	-	0.25	-	0.1	-	-	-	0.5
			VPEAR	10	0.5	-	2.6	1	-	-	-	0.5
9	5	252	Boost	-	-	0.25	-	0.1	-	-	-	0.5
			VPEAR	10	0.5	-	2.6	1	-	-		0.5
10	5	252	Boost	-	-	-	-	-	0.25	-	0.1	0.5
			VPEAR	10	0.5	-	-	-	-	2.6	1	0.5
**Abbreviations**	DEAE-D—diethylaminoethyl-dextran						GnRH—gonadotropin-releasing hormone					
	KLH—keyhole limpet hemocyanin						OVA—ovalbumin					
	SP—scrambled peptide						OTC—oxytetracycline					
	VPEAR—vaccine platform for extended antigen release											

## Data Availability

Data available on request due to privacy concerns relating to live animal use.

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
