# Peer review of "Delivering an Immunocastration Vaccine via a Novel Subcutaneous Implant"

_animals, 2022, doi:10.3390/ani12192698_

Round 1
Reviewer 1 Report
This research evaluates the effectiveness of a new subcutaneous device for the prolonged release of a vaccine for bovine immunocastration. Multiple combinations of carrier protein-bound antigen and different adjuvant concentrations were tested in several experiments. The effectiveness of immunocastration was determined by measuring testosterone and some parameters of testicular development. While the work is interesting, there are several elements that need to be improved to better understand the results.
Materials and methods:
It would be appropriate to incorporate the information and characteristics of the study animals as a first point.
There is no adequate description of the vaccine design. Why were 2 carrier proteins used?. Can there be a different potency for each carrier?. Why is this order of components used in the device?. Is the antigen soluble or in microparticles?. How does the release of prime, boost and VPEAR occur?
blood sampling: has been described that testosterone levels in blood vary during the day. The samples were always taken at the same time? all animals were of the same age? how was the increase of testosterone over time, considering that at 6 months the animals start puberty?
the ELISA assay says streptavidin coated plates were used, GnRH was conjugated with avidin? is it an in-house lab assay?
Why was the temperature measured, are there results of that? was it used to measure safety? why every 14 days?
is it possible to incorporate photos of the difference in Testes histology?
treatments: table 2 is complex to understand, the legend explaining the abbreviations is missing. I suggest grouping some pilots with the same variables versus their respective controls. it would be convenient to include in the table the duration of each trial and which efficacy parameters were evaluated.
Why were these adjuvants used? They can produce different cellular or humoral immunity profiles.
results and discussion:
Both adjuvants appear to influence spermatogenesis, however, it appears to be dose dependent? Is it possible that in a longer study the differences between adjuvants will be smaller?
Figure 2 shows the difference in antibody levels when adjuvants are used, but they were only high up to day 90. How long should the antibodies be high with a technology like this?.
in figure 3 the difference between group 7 and 8 could be attributed to the use of 2 types of antigens?
the effect of different types of adjuvants on the duration of immunocastration was partly analyzed by these authors DOI: 10.1111/aji.12772.
Pilot 5 shown in Figure 4 does not seem to contain relevant information in relation to pilot 4.
The conclusions of the work are ambiguous and only highlight the increase in the levels of antibodies obtained.
Author Response
Please see the attachment.
We appreciate the helpful feedback and feel that the manuscript has been strengthened.

Reviewer 2 Report
The propects for applicabilty of this pilot study are very interesting. However, as this is a pilot study it should be clearly discussed as such. Therefore my main point of criticism is how the results on implant rejection are presented and discussed. It should be mentioned in the results part for all pilot studies and should be more than one sentence at the end of a discussion paragraph. If possible, a connection of the results of blood parameters to the presence of the implants in the animals should be drawn in the results section. Where possible, results on testosterone in combination with other measured parameters should be presented and discussed for all pilots. Please see the attached document for more detailed comments.

Author Response

(The authors gave the same response as above.)

Round 2
Reviewer 1 Report
I appreciate the improvements made in the text, I only suggest incorporating in the vaccine design section, the adjuvants used and their concentration (DEAE Deaxtran and QuilA).
It would also be very appropriate to incorporate the histological images in the text.
Author Response
Thank you for the helpful comments. Please see the attachment.
